# Harmonization of ICF Body Structures and ICD-11 Anatomic Detail: One foundation for multiple classifications

**Vincenzo Della Mea** [1,2], **Ann-Helene Almborg** [3,4], **Michela Martinuzzi** [5], **Samson W. Tu** [6,7], **Andrea Martinuzzi** [8,2] *

**1** Dept. of Mathematics, Computer Science and Physics, University of Udine, Udine, Italy, **2** Italian WHO-FIC Collaboration Center, Trieste, Italy, **3** National Board of Health and Welfare, Stockholm, Sweden, **4** Nordic WHO-FIC Collaboration Center, Oslo, Norway, **5** Sandwell & West Birmingham NHS Trust, Birmingham, United Kingdom, **6** Stanford Center for Biomedical Informatics Research, Stanford University, Stanford, CA, United States of America, **7** Stanford WHO-FIC Collaboration Center, Stanford, CA, United States of America, **8** Department of Neurorehabilitation, Conegliano Research Centre, RCCS Medea, Conegliano, Italy

☉ These authors contributed equally to this work.
\* andrea.martinuzzi@lanostrafamiglia.it

**Data Availability Statement:** All relevant data, including consolidated maps, are made available in SSOM format from the GitHub repository (https://

## Abstract

The Family of International Classifications of the World Health Organization (WHO-FIC) currently includes three reference classifications, namely International Classification of Diseases (ICD), International Classification of Functioning, Disability, and Health (ICF), and International Classification of Health Interventions (ICHI). Recently, the three classifications have been incorporated into a single WHO-FIC Foundation that serves as the repository of all concepts in the classifications. Each classification serves a specific classification need. However, they share some common concepts that are present, in different forms, in two or all of them. For the WHO-FIC Foundation to be a logically consistent repository without duplicates, these common concepts must be reconciled. One important set of shared concepts is the representation of human anatomy entities, which are not always modeled in the same way and with the same level of detail. To understand the relationships among the three anatomical representations, an effort is needed to compare them, identifying common areas, gaps, and compatible and incompatible modeling. The work presented here contributes to this effort, focusing on the anatomy representations in ICF and ICD-11. For this aim, three experts were asked to identify, for each entity in the ICF Body Structures, one or more entities in the ICD-11 Anatomic Detail that could be considered identical, broader or narrower. To do this, they used a specifically developed web application, which also automatically identified the most obvious equivalences. A total of 631 maps were independently identified by the three mappers for 218 ICF Body Structures, with an interobserver agreement of 93.5%. Together with 113 maps identified by the software, they were then consolidated into 434 relations. The results highlight some differences between the two classifications: in general, ICF is less detailed than ICD-11; ICF favors lumping of structures; in very few cases, the two classifications follow different anatomic models. For these issues, solutions have to be found that are compliant with the WHO approach to classification modeling and maintenance.

github.com/whoficitc/harmonization/blob/main/ICF-ICD-anatomy-v1.tsv).

**Funding:** Funder Name: Ministero della Salute (Italian Ministry of Health) Grant Number: RC2021-22 Grant Recipient: dr Andrea Martinuzzi The funders had no role in study design, data collection and analysis, decision to publish, or preparation of the manuscript.

**Competing interests:** The authors have declared that no competing interests exist.

## Introduction

The Family of International Classifications of the World Health Organization (WHO-FIC) currently includes three reference classifications, namely:

- ICD: the International Classification of Diseases

- ICF: the International Classification of Functioning, Disabilities and Health

- ICHI: the International Classification of Health Interventions

Each of the classifications covers a specific need that may arise when classifying the content of health documentation. Furthermore, while they share common principles, their structures depend in part on choices made at their initial design and development phases, which were done at different times and which in turn depend on knowledge and constraints defined at the times. In particular, the 11th revision of ICD [1] marks a departure from the paper-based model that led the development of the previous revisions, with a specific attention to information-technology aspects, including proper formalization of the represented concepts, and a double-layered structure composed of an ICD-11 Foundation that includes all possible ICD-related information and one or more *linearizations* that represents different versions of the classification designed to satisfy specific use cases [2]. The ICD-11 experience informs the development of ICHI, which is the youngest classification of the family; on the other side, ICF might be considered the oldest one, as it is still oriented towards traditional paper-based usage, although recently it is subjected to some enhancement in the computerized direction [3, 4].

The most notable recent enhancement to the three classifications is that, following the development of a new architecture for ICD-11 [5], all of them have been incorporated into a single WHO-FIC Foundation [6] which is an integrated semantic network consisting of all of the knowledge related to the concepts in the three classifications. Formalized in the Web Ontology Language (OWL), the Foundation organizes concepts in polyhierarchies that have no residual terms. The concepts are structured according to a WHO-FIC Content Model [7] that specifies the necessary attributes, such as titles and definitions, of a concept; its lexical properties, such as synonyms and inclusions; and the logical organization of the hierarchies, such as the allowed and required post-coordination axes. Different classifications, such as ICF, ICHI, and versions of ICD-11 designed for different use cases, can be generated automatically as different linearizations according to their specifications in the Foundation. The WHO-FIC Foundation promises to be a modern terminology system that is both backward compatible with the traditional requirements of statistical classifications and also prepared for linking and mapping to other terminologies and ontologies. Currently, in this combined representation, ICD-11 and ICF concepts are represented separately, while ICHI, because of its recent provenance, references some ICD-11 and ICF concepts.

For the WHO-FIC Foundation to realize its promise, it needs to reorganize the ICD-11, ICHI, and ICF concepts so that they form a logically consistent and semantically integrated and linked whole. Even though they cover different needs, the classifications share common concepts that are currently present in different forms in the Foundation. ICF, for example, includes body functions such as "Sensation of pain" (b280) that are closely related to ICD-11's pain symptoms. Furthermore, with the introduction of post-coordination [8], ICD-11 includes extension codes, such as anatomical entities and health devices, equipment and supplies, that are not in past versions of ICD and that increase the possibility of overlaps among the classifications. Identifying, characterizing, and harmonizing all the possible shared concepts is a huge and complex task. It is the aim of a current WHO effort to harmonize the shared concepts among the reference classifications of the family so that duplicates are removed and related

concepts are organized in consistent hierarchies. Such harmonized concepts across classifications will facilitate the joint use of the classifications, for example in documenting disability characteristics of children in early intervention [9] or determining functioning and disability profiles of chronic stroke patients [10]. ICD-11 codes post-coordinated with anatomical details and ICF-coded data on body structure impairments can be made interoperable just as ICHI health intervention codes already use ICF body functions and activities and participation codes as targets. Such consistent coding practices will facilitate coordinated care and enhance data integration and analysis. These benefits to users of the classifications and of the Foundation cannot be realized until this harmonization is completed.

Identifying, characterizing, and harmonizing all the possible shared concepts is a huge and complex task. Groups in the WHO-FIC community are embarking on projects to identify areas of overlap among the classifications, to understand the use cases for different types of harmonization, to investigate methods for mapping the related concepts, and to organize the necessary workflow and governance to propose, consider, and approve necessary changes to the Foundation and to the classifications. However there is one set of concepts that is obviously shared and easy to recognize, i.e., human anatomy. In fact, an anatomical entity may define the site of a disease in ICD-11, the body structure subject to impairment in ICF, and also the target of some health interventions defined in ICHI. Thus, the harmonization of the anatomical entities in the three classifications presents a useful case study that will inform the larger harmonization work.

While they are present everywhere, it is not obvious that all the anatomical entities are modeled the same way and with the same level of detail in the three reference classifications. To understand the relationships among the three anatomical representations, an effort is needed to compare them, identifying common areas, gaps, and compatible and incompatible models. The work presented here contributes to this effort, focusing on the anatomy representations in ICF and ICD-11 at first.

The aims of the present paper are thus:

- To identify anatomical detail as represented in ICF and ICD-11;

- To compare the representations by setting relationships among specific entities;

- To characterize the level of detail in each classification;

- To characterize areas where the underlying modeling is different or even incompatible;

- Finally, to suggest measures to be taken to harmonize and consolidate the representations.

While there exists a deeply formalized representation of anatomy, namely the Foundational Model of Anatomy [11], it is much too detailed for our goal, which is to harmonize the anatomical representations in the WHO-FIC Foundation, minimizing the changes needed to the classifications, with the aim of continuity and compatibility with current applications.

## Methods

To understand the relationship between ICF and ICD-11 anatomy, we decided to begin by mapping anatomical entities in the two classifications. In particular, the specific areas considered were the ICF Body Structures, and the section of ICD-11 Extension codes called "Anatomic Detail". Both have a partially heterogeneous hierarchical representation, including both partonomic and taxonomic relationships.

## Expert mappings

An initial qualitative analysis showed that inside these subsets, some entities have exactly the same title, some others are named differently, and some others do not have a direct correspondence, because an entity explicitly mentioned in one classification may only be found as part of a larger, or as specialization of a more general entity in the other classification.

Thus, three experts were asked to identify, for each entity in the ICF Body Structures, one equivalent entity in the ICD-11 Anatomic Detail if possible; if not, one or more entities that could be considered broader_than or narrower_than. We opted for these generic relationships because they encompass both the partonomic and taxonomic views. Furthermore, for practical aims, we also subdivided the equivalence relationship in two subtypes: identical_to, when the title and concept was substantially the same, and synonym_of, when the title of the concept was different, yet equivalent. Matching has been attempted not only on the terminal entities in the hierarchy, but also for all the higher-level entities.

While this effort is not aimed at mapping electronic health records, when possible, the principles described in the WHO paper on mapping have been respected [12].

## Classifications

The source list for ICF Body Structures was obtained from a ClaML representation of ICF version 2017 and included 321 entities. Of these 321, 103 were identified as residuals, i.e., categories of the kinds "not otherwise specified" or "other specified". Residuals were not mapped, because semantically equivalent to the parent category.

The target ICD-11 list was composed by a subset of the Extension Codes [13], namely the "Anatomical Detail" branch, taken from the Foundation layer. It was not extracted in advance, but dynamically obtained through the ICD-11 API [14] when needed, as explained in the next paragraphs.

## Software

To support the matching effort by the three experts, an ad-hoc web-based software has been developed. The main interface of the software shows the ICF Body Structures hierarchy on the left side, with the capability of selecting a kind of relationship among those previously mentioned, and the corresponding ICD-11 entity, for each ICF entity. The ICD-11 entity is identified by accessing an instance of the ICD-11 Foundation coding tool [14] on the right side, set to automatically search for the ICF title in the ICD-11 Foundation, Extension Codes chapter, "Anatomic Detail" branch. However, since this could not always be found, the expert may also edit the search term with synonyms that might be present in ICD-11. Fig 1 shows a screenshot from the Map Editor.

To reduce the effort needed, the software has a module that automatically identifies straightforward relationships, i.e., those where titles are identical, or obvious synonyms. Among the latter, many entities in ICF are named as "Structure of X", which were automatically mapped as synonyms of "X" if available in ICD-11. Technically "structure of X" is not synonymous with "X." However, given that parent/child links in both ICD-11 anatomy extension codes and ICF body structures are a mix of taxonomic and partonomic relationship, identifying "structure of X" with "X" yields the most economical set of maps. As an example, 'Structure of brain' from ICF is automatically mapped as synonym_of 'Brain' in ICD-11.

While doing their work, experts do not see matches already found by the others, in order to allow the computation of a measure of inter-observer agreement.

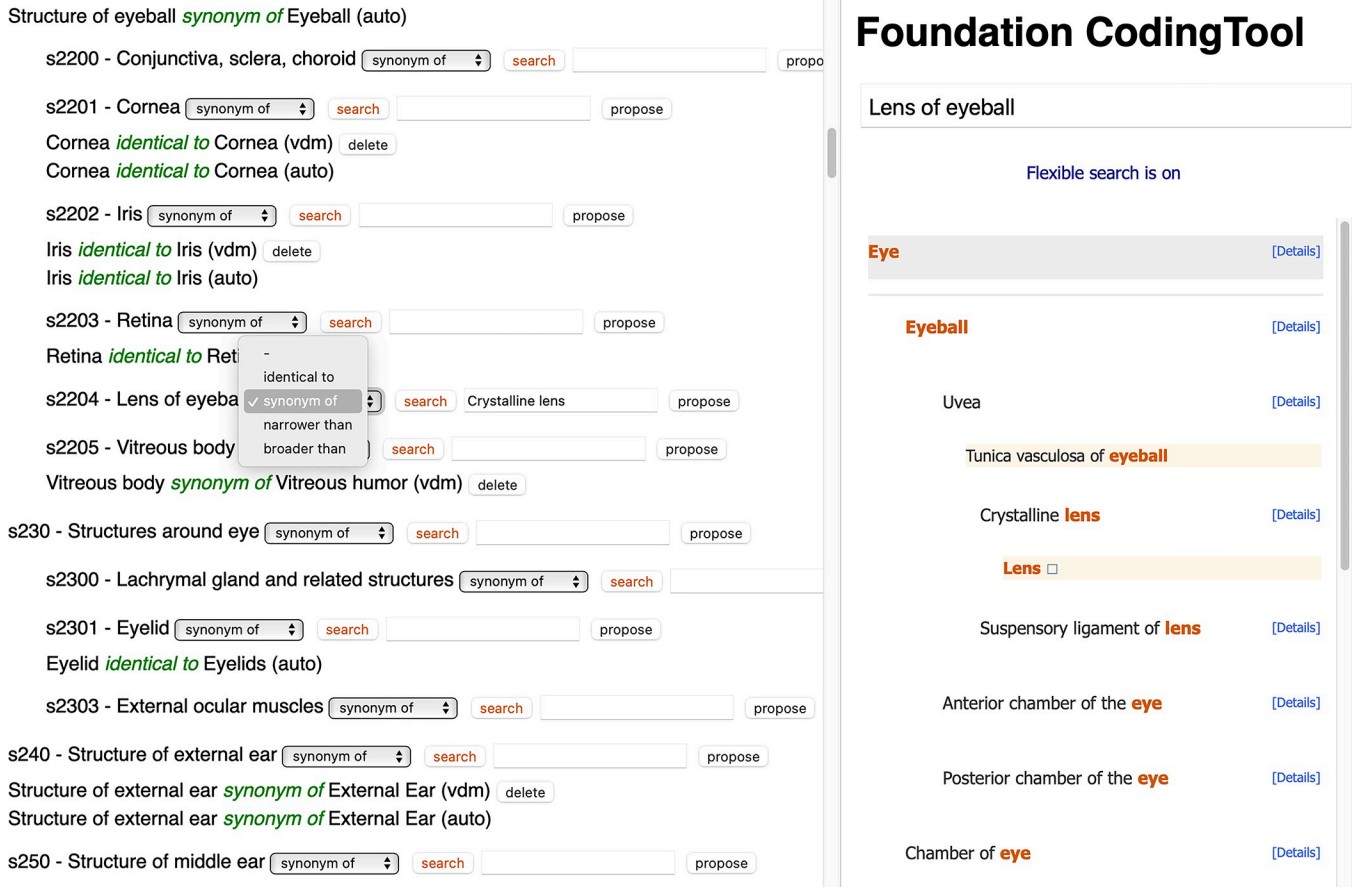

**Fig 1. On the left side, part of ICF Body Structures.** The user is selecting a relationship (synonym of) for the ICF entity "Lens of eyeball". On the right side, the ICD-11 Coding tool attempted a search with that term, which is not directly found. However, the expert may choose "Crystalline lens" and have it automatically inputed in the left side field, to propose it as candidate relationship. In the top left, some already defined relationships could be seen. Those identified as "auto" are automatically set by the software.

Another interface shows a tree representation of the ICF Body Structures, with color codings for the different relationships, when available. Fig 2 shows a screenshot depicting part of the ICF Body Structures tree with mappings.

Finally, a further module allows the export of matches to a CSV file for further processing.

## Results

Of 218 entities considered in the source list and excluding residuals, 113 were found by software to have identical corresponding entities in the ICD-11 Anatomic Detail extension, basing on lexical criteria.

Three independent knowledgeable raters considered the 105 entities for which the tool did not automatically confirm identity matches. The search for the most appropriate matching entity could result in finding identical, equivalent, broader or narrower items or no match. The three mappers independently produced 631 maps to a total of 297 ICD-11 entities, independently and without accessing others' maps until the end of the experiment. For all the ICF entities, at least one expert found a map in ICD-11. Table 1 shows a breakdown of the experts' mappings. While identical_to and synonym_of maps were collected, for the sake of simplicity,

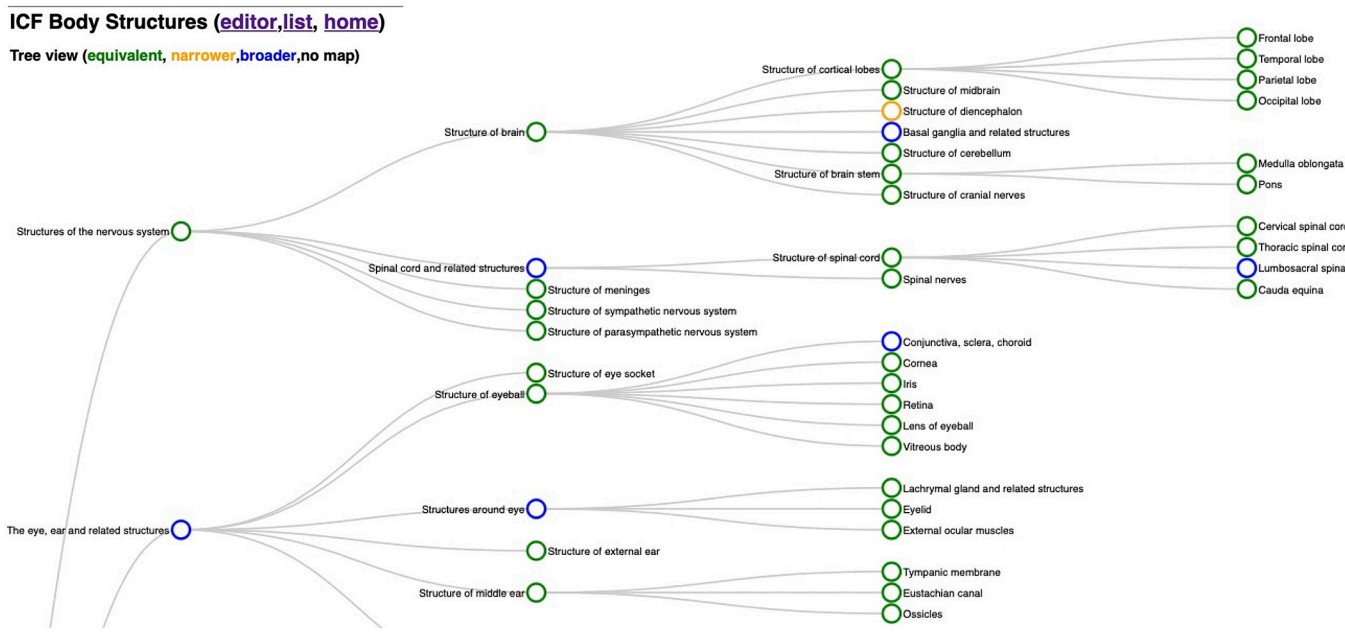

**Fig 2. Part of the tree view of the ICF Body Structure, with color-coded matches towards ICD-11.**

and considering that they are semantically the same, for the analysis they were collapsed in a single equivalent_to category.

## Interobserver agreement

Among the 26 in disagreement, some typical examples are as follows:

- Bronchial tree in ICF equivalent_to or broader_than Bronchus in ICD-11;

- Vaginal canal in ICF equivalent_to or narrower_than Vagina in ICD-11;

- Ligaments and fasciae of hand in ICF broader_than or narrower_than Ligament of the wrist and hand in ICD-11.

On the other side, by considering the 115 mapped ICF categories, some further differences became visible due to different choices in *narrower_than* or *broader_than* relationships (for example, obvious or redundant *broader_than* relationships). This involved the same ICF source, but possibly different ICD-11 targets. Examples from a total of 12 cases are:

- Structure of diencephalon narrower_than Supratentorial region of brain

  narrower_than Brain: this one is redundant
  broader_than Thalamus

**Table 1. Mappings proposed by each expert.**

| Mapper/Relations | equivalent_to | broader_than | narrower_than | total |
|---|---|---|---|---|
| A | 50 | 106 | 11 | 167 |
| B | 55 | 226 | 26 | 307 |
| C | 36 | 84 | 37 | 157 |
| **Total** | **141** | **416** | **74** | **631** |

**Table 2. Consolidated mappings.**

| Relations | *equivalent_to* | *broader_than* | *narrower_than* | total |
|---|---|---|---|---|
| Consolidated maps | 164 | 230 | 40 | 434 |

- Structure of skin glands broader_than Sebaceous gland

  broader_than Apocrine sweat gland
  broader_than Eccrine gland
  narrower_than Skin

- Atria equivalent_to Cardiac atrium

  broader_than Right atrium
  Discrepancies were examined by the experts to delete from the final set the maps on which there was not an agreement, that were redundant, or that were mistakes.

## Mapping details

The mappings have been consolidated in a single set of agreed-upon relationships, including automatically calculated equivalencies.

In order to obtain a consolidated mapping, we prioritized relationships as follows. First of all, if there is agreement on an equivalency, it is considered as the selected relationship, even if others are available. Without an equivalency, if a *narrower_than* relationship is available, that will be selected. Finally, *broader_than* relationships are selected.

After this step, the consolidated mappings are as described in Table 2.

## Discussion

The common Foundation from which the various WHO reference classifications are derived through linearizations requires that every entity be uniquely and unambiguously defined. This might require a process of harmonization of concepts that share similar meanings but may or may not be identical, also to favor joint use of the reference classifications. The process may be more or less complex according to the entities considered but could be easier for entities already sharing strong similarities. For further automatic mappings of concepts that may not be of the same semantic types, lexical mappings resulting in identical maps need to be evaluated to make sure that the concepts are identical. The "low hanging fruits" of harmonization include the anatomy entities as described in ICD-11 and in the body structure domain of ICF, which have been the subject of this paper.

Three main issues arise from this experiment are as follows:

a. Different levels of specificity: in general, ICF is less detailed than ICD (74 narrower_than vs 416 broader_thanmaps);

b. ICF favors lumping of structures: e.g. "The eye, ear and related structures"; "Structures of cardiovascular, immunological and respiratory systems"; "Structure of vagina and external genitalia"; "Testes and scrotum";

c. The two classifications may have different anatomic models: e.g., Where is the shoulder? For ICD: in upper extremity; for ICF: in structures related to movement.

For these issues, solutions have to be found that are compliant with WHO's new approach to classification modeling and maintenance, which includes the concept of a common Foundation that integrates all the relevant entities in a coherent terminological system based on

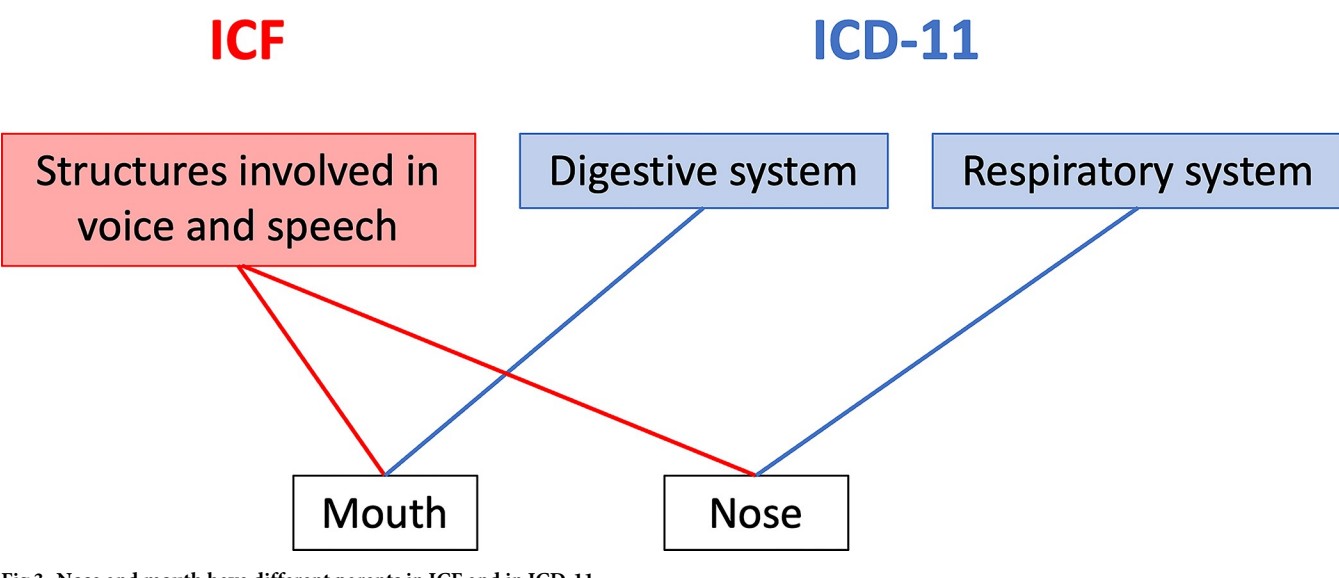

**Fig 3. Nose and mouth have different parents in ICF and in ICD-11.**

semantic web technology. Versions of traditional classifications are generated from the Foundation as linearizations.

Issue A does not represent a real problem: according to the approach for creating linearizations from the Foundation, the "boundary" of specificity is called "shoreline"; and to select different specificities, it is only a matter of defining a different shoreline for each classification. Thus, ICD-11's shoreline for anatomical concepts will include more detailed concepts than that of ICF.

Issue B might have two different solutions. The solution with least impact, from the point of view of classifications, is to introduce the ICF groupings as new groupings in the Foundation's Anatomic Detail, while not using them in the ICD-11 linearization. However, such groupings are often heterogeneous and missing a strong semantic similarity (e.g., "The eye, ear and related structures"), and are present only for the sake of aggregation in a classification that was born with a printed edition in mind, but can be useful for aggregation of data at a higher level. Thus, the ideal solution would be to revise the hierarchy organization in ICF, without modifying terminal entities, in order to split heterogeneous groupings into more semantically homogeneous groups.

Issue C is apparently the most complex: different models may mean a totally different hierarchy, and such differences in modeling is intrinsic in the different points of view expressed by ICF and ICD (the former centered on functioning, the latter on clinical aspects). Fig 3 shows a clear and compelling example: in ICF terms, mouth and nose are structures devoted to speech, while in ICD terms, mouth is part of the digestive system, and nose of the respiratory system. Both views are perfectly reasonable, yet both miss the aspect evidentiated by the other classification. Nevertheless, both views may be accommodated in the WHO-FIC Foundation, which differs from traditional classifications in that it allows for multiple parenthood relationships. The use of multiple parents might enable us to express the different views without forcing one classification to adopt the point of view of the other and leaving to the linearization phase the choice of one or the other parent in each classification.

This work makes three-fold contributions: first, it presents a principled exemplar of an approach to design and conduct terminological mappings; second, it uncovers specific commonalities and discrepancies between the anatomical concepts in ICD-11 and ICF and

proposes methods to harmonize them in the context of the WHO-FIC Foundation; third, it pioneered the tools and methods, and provides the first results of a larger program of work to map, integrate, and link concepts in the three core WHO-FIC classifications. Future work will investigate the relationships such as those between ICHI's anatomical targets and ICD-11's Anatomical Detail extension codes, between impairments of ICF body functions and ICD-11 signs and symptoms, and between ICF activities and participations and ICD-11 dimensions of external causes, among others. Just as we developed the web-based software to assist subject matter experts performing the mappings, we will investigate additional automated methods to map related concepts, possibly leveraging previously validated mappings as training sets.

The consolidated maps are made available in SSOM format [15] at https://github.com/whoficitc/harmonization/blob/main/ICF-ICD-anatomy-v1.tsv.

## Conclusion

The mapping of anatomical concepts in the ICF body structures domains and ICD-11 Anatomy and Topography extension codes yields insights into how the two classifications treat the representation of anatomy structures differently. We present methods for reconciling the differences. The case study points to the way additional work harmonizing the concepts in ICD-11, ICF, and ICHI may be conducted.

## Acknowledgments

The support of the Italian Ministry of Health to AM (RC2021-22-23) is gratefully acknowledged.

## Author Contributions

**Conceptualization:** Vincenzo Della Mea, Samson W. Tu, Andrea Martinuzzi.

**Data curation:** Vincenzo Della Mea, Ann-Helene Almborg, Michela Martinuzzi, Samson W. Tu, Andrea Martinuzzi.

**Formal analysis:** Vincenzo Della Mea, Ann-Helene Almborg, Samson W. Tu, Andrea Martinuzzi.

**Funding acquisition:** Andrea Martinuzzi.

**Investigation:** Vincenzo Della Mea, Ann-Helene Almborg, Michela Martinuzzi, Samson W. Tu, Andrea Martinuzzi.

**Methodology:** Vincenzo Della Mea.

**Validation:** Andrea Martinuzzi.

**Writing – original draft:** Vincenzo Della Mea, Andrea Martinuzzi.

**Writing – review & editing:** Ann-Helene Almborg, Michela Martinuzzi, Samson W. Tu.

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
