## [Decision Letter · Decision Letter 0]

16 Jan 2023

PONE-D-22-34808Harmonization of ICF Body Structures and ICD-11 Anatomic Detail: one foundation for two classificationsPLOS ONE

Dear Dr. Martinuzzi,

Thank you for submitting your manuscript to PLOS ONE. After careful consideration, we feel that it has merit but does not fully meet PLOS ONE’s publication criteria as it currently stands. Therefore, we invite you to submit a revised version of the manuscript that addresses the points raised during the review process.

We look forward to receiving your revised manuscript.

Kind regards,

Ernesto Iadanza

Academic Editor

PLOS ONE

Reviewers' comments:

Reviewer's Responses to Questions

**Comments to the Author**

1. Is the manuscript technically sound, and do the data support the conclusions?

Reviewer #1: Yes

Reviewer #2: No

2. Has the statistical analysis been performed appropriately and rigorously? 

Reviewer #1: I Don't Know

Reviewer #2: N/A

3. Have the authors made all data underlying the findings in their manuscript fully available?

Reviewer #1: Yes

Reviewer #2: Yes

4. Is the manuscript presented in an intelligible fashion and written in standard English?

Reviewer #1: Yes

Reviewer #2: Yes

5. Review Comments to the Author

Reviewer #1: It's always a useful step to begin creating such links across classification systems, and this appears to be a responsible, exploratory effort to begin building those bridges of equivalence and similarity where existing terminologies and hierarchies permit. However, this represents an admittedly small subset (434 relations) of all the complex and dnamic relationships that exist between the FIC terminologies. ICF has 1400 categories, and ICD-11 has 17,000 unique codes and more than 120,000 codable terms. If we consider the varieties of co-morbid conditions where multiple anatomical, physiological, and interventional factors may interact, our models will require both broader and more precise classifications. It's not clear to me what clinical advantage is gained by mapping 434 relations, in the midst of so many other un-mapped relations. It would be helpful in the Abstract to specify what specific scientific or clinical benefits can be expected from this improved cross-classification mapping.

I'm also a bit concerned about the accepted constraints implicit in the harmonization initiative, which prioritize compliance "with the WHO approach to classification modeling and maintenance." This risks constraining future terminologies and relationships within pre-existing frameworks which may simply be, or become, outdated or too crude to represent emerging complexities of scientific discoveries (e.g., chemotherapy vs. bioelectric field therapeutics). The "least common denominator" of harmony may end up giving us classification models that are less usefully precise.

Finally, I think we need to consider the growing relevance of semantic web data science and graph knowledge-base technologies as relevant platforms for making the vast scope of medical classifications and individual patient informatics more dynamically evolvable, make research more precisely and rapidly searchable, and make health informatics more interoperable. WHO's operational conservatism in classification may be locking the world of clinical research and services into obsolescent relational database frameworks of classification that will increasingly not mirror the dynamism and complex interdependencies of human health, Social Determinants of Health, clinical sciences and technological innovation.

Reviewer #2: The authors describe the three WHO classification systems (ICD-11, ICF, ICHI) and their different usage. Developed a user-friendly interface for coders to map the ICD-11 with ICF. Elaborated on the detail different between ICD-11 and ICF in anatomy entities.

The article needs to be revised majorly for publication:

1. Describe the usage of this harmonization? Where and under what situation will the scientists or healthcare workers/administrators use this?

2. Consider artificial intelligence methods. Use the experts’ results as training data for developing a model to reduce the experts’ efforts.

3. Map ICD-11, ICF and ICHI.

4. Supplement the Conclusion part.

5. Describe the innovation of this research.

6. PLOS authors have the option to publish the peer review history of their article (what does this mean?). If published, this will include your full peer review and any attached files.

Reviewer #1: No

Reviewer #2: **Yes: **Qiong Wang

---

## [Decision Letter · Decision Letter 1]

14 Mar 2023

PONE-D-22-34808R1Harmonization of ICF Body Structures and ICD-11 Anatomic Detail: one foundation for multiple classificationsPLOS ONE

Dear Dr. Martinuzzi,

Thank you for submitting your manuscript to PLOS ONE. After careful consideration, we feel that it has merit but does not fully meet PLOS ONE’s publication criteria as it currently stands. Therefore, we invite you to submit a revised version of the manuscript that addresses the points raised during the review process.

We look forward to receiving your revised manuscript.

Kind regards,

Ernesto Iadanza

Academic Editor

PLOS ONE

Journal Requirements:

Reviewers' comments:

Reviewer's Responses to Questions

**Comments to the Author**

1. If the authors have adequately addressed your comments raised in a previous round of review and you feel that this manuscript is now acceptable for publication, you may indicate that here to bypass the “Comments to the Author” section, enter your conflict of interest statement in the “Confidential to Editor” section, and submit your "Accept" recommendation.

Reviewer #2: All comments have been addressed

2. Is the manuscript technically sound, and do the data support the conclusions?

Reviewer #2: Yes

3. Has the statistical analysis been performed appropriately and rigorously? 

Reviewer #2: Yes

4. Have the authors made all data underlying the findings in their manuscript fully available?

Reviewer #2: No

5. Is the manuscript presented in an intelligible fashion and written in standard English?

Reviewer #2: Yes

6. Review Comments to the Author

Reviewer #2: I deeply appreciate the authors' efforts in advancing the WHO-FIC through their work. It’s always useful to start from the coding step to try to manage and utilize healthcare data. But the author did not expound on the significance of their work. The WHO-FIC Foundation initially comprised the International Classification of Diseases (ICD), International Classification of Functioning, Disability and Health (ICF), and International Classification of Health Interventions (ICHI), which were designed to categorize distinct domains of health information. From a statistical and data analysis perspective, it is unclear how their work will be used in the future. In terms of medical terminology, pairing ICD and ICF appears to conflict with the primary aim of separating different fields of classification. It is worth noting that the citation does not indicate that the WHO-FIC Foundation intends to amalgamate the three coding systems.

Based on my understanding, the involvement of human interaction in the development of natural language processing (NLP) methods for healthcare is critical but can be both time and cost-intensive. If the authors make their data publicly available, it could prove to be a valuable resource for training NLP methods. In this regard, I would encourage the authors to present their matching results in such a way that they could serve as golden standards for algorithm training, particularly with regard to entity recognition.

7. PLOS authors have the option to publish the peer review history of their article (what does this mean?). If published, this will include your full peer review and any attached files.

Reviewer #2: **Yes: **Qiong Wang

While revising your submission, please upload your figure files to the Preflight Analysis and Conversion Engine (PACE) digital diagnostic tool, https://pacev2.apexcovantage.com/. PACE helps ensure that figures meet PLOS requirements. To use PACE, you must first register as a user. Registration is free. Then, login and navigate to the UPLOAD tab, where you will find detailed instructions on how to use the tool. If you encounter any issues or have any questions when using PACE, please email PLOS at figures@plos.org. Please note that Supporting Information files do not need this step.<quillbot-extension-portal></quillbot-extension-portal>

---

## [Author Response · Author response to Decision Letter 1]

31 Mar 2023

see the attached document and also below: 

Reviewer #2: I deeply appreciate the authors' efforts in advancing the WHO-FIC through their work. It’s always useful to start from the coding step to try to manage and utilize healthcare data. But the author did not expound on the significance of their work. The WHO-FIC Foundation initially comprised the International Classification of Diseases (ICD), International Classification of Functioning, Disability and Health (ICF), and International Classification of Health Interventions (ICHI), which were designed to categorize distinct domains of health information. From a statistical and data analysis perspective, it is unclear how their work will be used in the future. In terms of medical terminology, pairing ICD and ICF appears to conflict with the primary aim of separating different fields of classification. It is worth noting that the citation does not indicate that the WHO-FIC Foundation intends to amalgamate the three coding systems.

>> Response

ICD, ICF, and ICHI, while each focusing on its specific health and health care domains, are designed to complement each other, not artificially separated. There are already closely related concepts in the classifications, for example, between ICF’s pain body functions and sensation and ICD-11’s pain symptoms. Especially with the introduction of extension codes in ICD-11, there are significant overlaps in the coverage of the three classifications. With joint use of the classifications, collected data need to be consistent (e.g., ICD-11 codes with anatomical details and ICF body structure impairment). Consistent coding practices in joint use of the classifications will facilitate coordinated care and enhance data integration and analysis. In the revised paragraph in the Introduction, we make these points explicitly and cite a couple examples of the joint use of ICD and ICF.

For the WHO-FIC Foundation to realize its promise, it needs to reorganize the ICD-11, ICHI, and ICF concepts so that they form a logically consistent and semantically integrated and linked whole. Even though they cover different needs, the classifications share common concepts that are currently present in different forms in the Foundation. ICF, for example, includes body functions such as “Sensation of pain” (b280) that are closely related to ICD-11’s pain symptoms. Furthermore, with the introduction of post-coordination (8), ICD-11 includes extension codes, such as anatomical entities and health devices, equipment, and supplies, that are not in past versions of ICD and that increase the possibility of overlaps among the classifications. Identifying, characterizing, and harmonizing all the possible shared concepts is a huge and complex task. It is the aim of a current WHO effort to harmonize the shared concepts among the reference classifications of the family so that duplicates are removed and related concepts are organized in consistent hierarchies. Such harmonized concepts across classifications will facilitate the joint use of the classifications, for example in documenting disability characteristics of children in early intervention (9) or determining functioning and disability profiles of chronic stroke patients (10). ICD-11 codes post-coordinated with anatomical details and ICF-coded data on body structure impairments can be made interoperable just as ICHI health intervention codes already use ICF body functions and activities and participation codes as targets. Such consistent coding practices will facilitate coordinated care and enhance data integration and analysis. These benefits to users of the classifications and of the Foundation cannot be realized until this harmonization is completed.

(8) Mabon K, Steinum O, Chute CG. Postcoordination of codes in ICD-11. BMC Med Inform Decis Mak. 2022;21(Suppl 6):379.

(9) Martins EF, de Sousa PH, Barbosa PH, de Menezes LT, Costa AS. A Brazilian experience to describe functioning and disability profiles provided by combined use of ICD and ICF in chronic stroke patients at home-care. Disabil Rehabil. 2011;33(21-22):2064-74.

(10) Simeonsson RJ, Scarborough AA, Hebbeler KM. ICF and ICD codes provide a standard language of disability in young children. J Clin Epidemiol. 2006;59(4):365-73.

We are surprised that the reviewer said "the citation does not indicate that the WHO-FIC Foundation intends to amalgamate the three coding systems.” While the integration of the three classification in a single WHO-FIC Foundation is work in progress and is not yet described in journal publications, the Foundation reference (#6 in the reviewed submission) points to a web page that clearly shows that the three classifications are in the Foundation. If the reviewer’s point is that the three classifications will continue to exist as separate classifications for coding, that is true, but misses the point that they should be generated as linearizations from the single WHO-FIC Foundation where all of the classification concepts are amalgamated and made consistent.

>> Reviewer’s comment

Based on my understanding, the involvement of human interaction in the development of natural language processing (NLP) methods for healthcare is critical but can be both time and cost-intensive. If the authors make their data publicly available, it could prove to be a valuable resource for training NLP methods. In this regard, I would encourage the authors to present their matching results in such a way that they could serve as golden standards for algorithm training, particularly with regard to entity recognition.

>> Response

As stated in the reviewed manuscript "The consolidated maps are made available in SSOM format (12) at https://github.com/whoficitc/harmonization/blob/main/ICF-ICD-anatomy-v1.tsv.” SSOM format is a simple tabular structure that is easy for anyone who wants to re-use the data to reformat for their purpose.

---

## [Editor Report · Decision Letter 2]

10 Apr 2023

Harmonization of ICF Body Structures and ICD-11 Anatomic Detail: one foundation for multiple classifications

PONE-D-22-34808R2

Dear Dr. Martinuzzi,

We’re pleased to inform you that your manuscript has been judged scientifically suitable for publication and will be formally accepted for publication once it meets all outstanding technical requirements.

Kind regards,

Ernesto Iadanza

Academic Editor

PLOS ONE

Additional Editor Comments (optional):

Reviewers' comments:

<quillbot-extension-portal></quillbot-extension-portal>

---

## [Editor Report · Acceptance letter]

13 Apr 2023

PONE-D-22-34808R2 

Harmonization of ICF Body Structures and ICD-11 Anatomic Detail: One Foundation for Multiple Classifications 

Dear Dr. Martinuzzi:

I'm pleased to inform you that your manuscript has been deemed suitable for publication in PLOS ONE. Congratulations! Your manuscript is now with our production department. 

Kind regards, 

on behalf of

Dr. Ernesto Iadanza 

Academic Editor

PLOS ONE